# Differentiable Learning of Logical Rules for Knowledge Base Reasoning

**Fan Yang**      **Zhilin Yang**      **William W. Cohen**
School of Computer Science
Carnegie Mellon University
{fanyang1,zhiliny,wcohen}@cs.cmu.edu

## Abstract

We study the problem of learning probabilistic first-order logical rules for knowledge base reasoning. This learning problem is difficult because it requires learning the parameters in a continuous space as well as the structure in a discrete space. We propose a framework, *Neural Logic Programming*, that combines the parameter and structure learning of first-order logical rules in an end-to-end differentiable model. This approach is inspired by a recently-developed differentiable logic called TensorLog [5], where inference tasks can be compiled into sequences of differentiable operations. We design a neural controller system that learns to compose these operations. Empirically, our method outperforms prior work on multiple knowledge base benchmark datasets, including Freebase and WikiMovies.

## 1 Introduction

A large body of work in AI and machine learning has considered the problem of learning models composed of sets of first-order logical rules. An example of such rules is shown in Figure 1. Logical rules are useful representations for knowledge base reasoning tasks because they are interpretable, which can provide insight to inference results. In many cases this interpretability leads to robustness in transfer tasks. For example, consider the scenario in Figure 1. If new facts about more companies or locations are added to the knowledge base, the rule about `HasOfficeInCountry` will still be usefully accurate without retraining. The same might not be true for methods that learn embeddings for specific knowledge base entities, as is done in TransE [3].

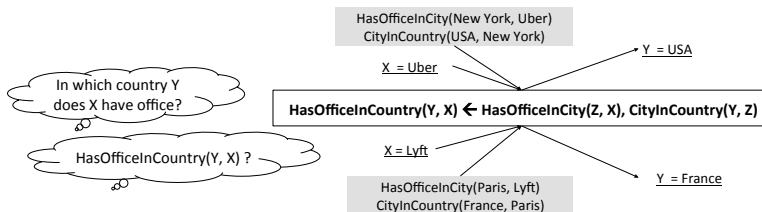

Figure 1: Using logical rules (shown in the box) for knowledge base reasoning.

Learning collections of relational rules is a type of *statistical relational learning* [7], and when the learning involves proposing new logical rules, it is often called *inductive logic programming* [18]. Often the underlying logic is a probabilistic logic, such as Markov Logic Networks [22] or ProPPR [26]. The advantage of using a probabilistic logic is that by equipping logical rules with probability, one can better model statistically complex and noisy data. Unfortunately, this learning problem is quite difficult — it requires learning both the *structure* (i.e. the particular sets of rules included in a model) and the *parameters* (i.e. confidence associated with each rule). Determining

the structure is a *discrete* optimization problem, and one that involves search over a potentially large problem space. Many past learning systems have thus used optimization methods that interleave moves in a discrete structure space with moves in parameter space [12, 13, 14, 27].

In this paper, we explore an alternative approach: a completely *differentiable* system for learning models defined by sets of first-order rules. This allows one to use modern gradient-based programming frameworks and optimization methods for the inductive logic programming task. Our approach is inspired by a differentiable probabilistic logic called TensorLog [5]. TensorLog establishes a connection between inference using first-order rules and sparse matrix multiplication, which enables certain types of logical inference tasks to be compiled into sequences of differentiable numerical operations on matrices. However, TensorLog is limited as a learning system because it only learns parameters, not rules. In order to learn parameters and structure simultaneously in a differentiable framework, we design a neural controller system with an attention mechanism and memory to *learn* to sequentially compose the primitive differentiable operations used by TensorLog. At each stage of the computation, the controller uses attention to "softly" choose a subset of TensorLog's operations, and then performs the operations with contents selected from the memory. We call our approach *neural logic programming*, or Neural LP.

Experimentally, we show that Neural LP performs well on a number of tasks. It improves the performance in knowledge base completion on several benchmark datasets, such as WordNet18 and Freebase15K [3]. And it obtains state-of-the-art performance on Freebase15KSelected [25], a recent and more challenging variant of Freebase15K. Neural LP also performs well on standard benchmark datasets for statistical relational learning, including datasets about biomedicine and kinship relationships [12]. Since good performance on many of these datasets can be obtained using short rules, we also evaluate Neural LP on a synthetic task which requires longer rules. Finally, we show that Neural LP can perform well in answering *partially structured* queries, where the query is posed partially in natural language. In particular, Neural LP also obtains state-of-the-art results on the KB version of the WIKIMOVIES dataset [16] for question-answering against a knowledge base. In addition, we show that logical rules can be recovered by executing the learned controller on examples and tracking the attention.

To summarize, the contributions of this paper include the following. First, we describe Neural LP, which is, to our knowledge, the first end-to-end differentiable approach to learning not only the parameters but also the structure of logical rules. Second, we experimentally evaluate Neural LP on several types of knowledge base reasoning tasks, illustrating that this new approach to inductive logic programming outperforms prior work. Third, we illustrate techniques for visualizing a Neural LP model as logical rules.

## 2 Related work

Structure embedding [3, 24, 29] has been a popular approach to reasoning with a knowledge base. This approach usually learns a embedding that maps knowledge base relations (e.g `CityInCountry`) and entities (e.g. `USA`) to tensors or vectors in latent feature spaces. Though our Neural LP system can be used for similar tasks as structure embedding, the methods are quite different. Structure embedding focuses on learning representations of relations and entities, while Neural LP learns logical rules. In addition, logical rules learned by Neural LP can be applied to entities not seen at training time. This is not achievable by structure embedding, since its reasoning ability relies on entity-dependent representations.

Neural LP differs from prior work on logical rule learning in that the system is end-to-end differentiable, thus enabling gradient based optimization, while most prior work involves discrete search in the problem space. For instance, Kok and Domingos [12] interleave beam search, using discrete operators to alter a rule set, with parameter learning via numeric methods for rule confidences. Lao and Cohen [13] introduce all rules from a restricted set, then use lasso-style regression to select a subset of predictive rules. Wang et al. [27] use an Iterative Structural Gradient algorithm that alternate gradient-based search for parameters of a probabilistic logic ProPPR [26], with structural additions suggested by the parameter gradients.

Recent work on neural program induction [21, 20, 1, 8] have used attention mechanism to "softly choose" differentiable operators, where the attentions are simply approximations to binary choices. The main difference in our work is that attentions are treated as *confidences* of the logical rules and

have semantic meanings. In other words, Neural LP learns a distribution over logical rules, instead of an approximation to a particular rule. Therefore, we do not use hardmax to replace softmax during inference time.

## 3 Framework

### 3.1 Knowledge base reasoning

Knowledge bases are collections of relational data of the format `Relation(head,tail)`, where `head` and `tail` are entities and `Relation` is a binary relation between entities. Examples of such data tuple are `HasOfficeInCity(New York,Uber)` and `CityInCountry(USA,New York)`.

The knowledge base reasoning task we consider here consists of a `query`[1], an entity `tail` that the `query` is about, and an entity `head` that is the answer to the query. The goal is to retrieve a ranked list of entities based on the `query` such that the desired answer (i.e. `head`) is ranked as high as possible.

To reason over knowledge base, for each `query` we are interested in learning weighted chain-like logical rules of the following form, similar to stochastic logic programs [19],

$$\alpha \; \texttt{query(Y,X)} \leftarrow \texttt{R_n(Y,Z_n)} \wedge \cdots \wedge \texttt{R_1(Z_1,X)} \tag{1}$$

where $\alpha \in [0,1]$ is the *confidence* associated with this rule, and $\texttt{R}_1, \ldots, \texttt{R}_n$ are relations in the knowledge base. During inference, given an entity `x`, the *score* of each `y` is defined as sum of the confidence of rules that imply `query(y,x)`, and we will return a ranked list of entities where higher the score implies higher the ranking.

### 3.2 TensorLog for KB reasoning

We next introduce TensorLog operators and then describe how they can be used for KB reasoning. Given a knowledge base, let $\mathbf{E}$ be the set of all entities and $\mathbf{R}$ be the set of all binary relations. We map all entities to integers, and each entity `i` is associated with a one-hot encoded vector $\mathbf{v}_i \in \{0,1\}^{|\mathbf{E}|}$ such that only the i-th entry is 1. TensorLog defines an operator $\mathbf{M}_\texttt{R}$ for each relation `R`. Concretely, $\mathbf{M}_\texttt{R}$ is a matrix in $\{0,1\}^{|\mathbf{E}| \times |\mathbf{E}|}$ such that its $(i,j)$ entry is 1 if and only if `R(i,j)` is in the knowledge base, where `i` is the i-th entity and similarly for `j`.

We now draw the connection between TensorLog operations and a restricted case of logical rule inference. Using the operators described above, we can imitate logical rule inference `R(Y,X) ← P(Y,Z) ∧ Q(Z,X)` for any entity `X = x` by performing matrix multiplications $\mathbf{M}_\texttt{P} \cdot \mathbf{M}_\texttt{Q} \cdot \mathbf{v}_\texttt{x} \doteq \mathbf{s}$. In other words, the non-zero entries of the vector $\mathbf{s}$ equals the set of `y` such that there exists `z` that `P(y,z)` and `Q(z,x)` are in the KB. Though we describe the case where rule length is two, it is straightforward to generalize this connection to rules of any length.

Using TensorLog operations, what we want to learn for each `query` is shown in Equation 2,

$$\sum_l \alpha_l \Pi_{\texttt{k} \in \beta_l} \mathbf{M}_{\texttt{R_k}} \tag{2}$$

where $l$ indexes over all possible rules, $\alpha_l$ is the confidence associated with rule $l$ and $\beta_l$ is an ordered list of all relations in this particular rule. During inference, given an entity $\mathbf{v}_\texttt{x}$, the *score* of each retrieved entity is then equivalent to the entries in the vector $\mathbf{s}$, as shown in Equation 3.

$$\mathbf{s} = \sum_l \left( \alpha_l \left( \Pi_{\texttt{k} \in \beta_l} \mathbf{M}_{\texttt{R_k}} \mathbf{v}_\texttt{x} \right) \right), \; \text{score}(\texttt{y} \mid \texttt{x}) = \mathbf{v}_\texttt{y}^T \mathbf{s} \tag{3}$$

To summarize, we are interested in the following learning problem for each `query`.

$$\max_{\{\alpha_l, \beta_l\}} \sum_{\{\texttt{x},\texttt{y}\}} \text{score}(\texttt{y} \mid \texttt{x}) = \max_{\{\alpha_l, \beta_l\}} \sum_{\{\texttt{x},\texttt{y}\}} \mathbf{v}_\texttt{y}^T \left( \sum_l \left( \alpha_l \left( \Pi_{\texttt{k} \in \beta_l} \mathbf{M}_{\texttt{R_k}} \mathbf{v}_\texttt{x} \right) \right) \right) \tag{4}$$

where $\{\texttt{x},\texttt{y}\}$ are entity pairs that satisfy the `query`, and $\{\alpha_l, \beta_l\}$ are to be learned.

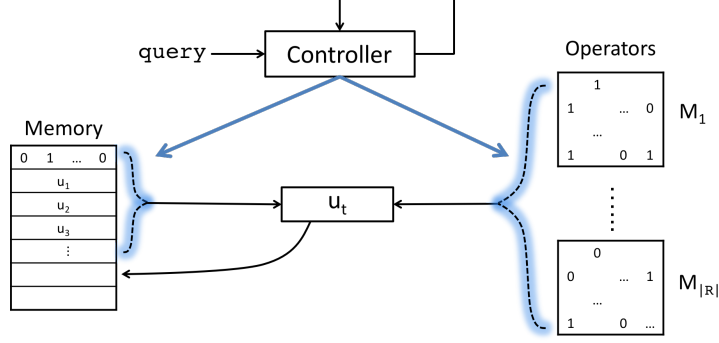

Figure 2: The neural controller system.

### 3.3 Learning the logical rules

We will now describe the *differentiable* rule learning process, including learnable parameters and the model architecture. As shown in Equation 2, for each query, we need to learn the set of rules that imply it and the confidences associated with these rules. However, it is difficult to formulate a *differentiable* process to directly learn the parameters and the structure $\{\alpha_l, \beta_l\}$. This is because each parameter is associated with a particular rule, and enumerating rules is an inherently discrete task. To overcome this difficulty, we observe that a different way to write Equation 2 is to interchange the summation and product, resulting the following formula with a different parameterization,

$$\prod_{t=1}^{T} \sum_{k}^{|\mathbf{R}|} a_t^k \mathbf{M}_{R_k} \tag{5}$$

where $T$ is the max length of rules and $|\mathbf{R}|$ is the number of relations in the knowledge base. The key parameterization difference between Equation 2 and Equation 5 is that in the latter we associate each relation in the rule with a weight. This combines the rule enumeration and confidence assignment.

However, the parameterization in Equation 5 is not sufficiently expressive, as it assumes that all rules are of the same length. We address this limitation in Equation 6-8, where we introduce a recurrent formulation similar to Equation 3.

In the recurrent formulation, we use auxiliary *memory vectors* $\mathbf{u}_t$. Initially the memory vector is set to the given entity $\mathbf{v_x}$. At each step as described in Equation 7, the model first computes a weighted average of previous memory vectors using the *memory attention vector* $\mathbf{b_t}$. Then the model "softly" applies the TensorLog operators using the *operator attention vector* $\mathbf{a_t}$. This formulation allows the model to apply the TensorLog operators on all previous partial inference results, instead of just the last step's.

$$\mathbf{u_0} = \mathbf{v_x} \tag{6}$$

$$\mathbf{u_t} = \sum_{k}^{|\mathbf{R}|} a_t^k \mathbf{M}_{R_k} \left( \sum_{\tau=0}^{t-1} b_t^\tau \mathbf{u}_\tau \right) \quad \text{for } 1 \leq t \leq T \tag{7}$$

$$\mathbf{u_{T+1}} = \sum_{\tau=0}^{T} b_{T+1}^\tau \mathbf{u}_\tau \tag{8}$$

Finally, the model computes a weighted average of all memory vectors, thus using attention to select the proper rule length. Given the above recurrent formulation, the learnable parameters for each query are $\{\mathbf{a_t} \mid 1 \leq t \leq T\}$ and $\{\mathbf{b_t} \mid 1 \leq t \leq T+1\}$.

We now describe a neural controller system to learn the operator and memory attention vectors. We use recurrent neural networks not only because they fit with our recurrent formulation, but also because it is likely that current step's attentions are dependent on previous steps'. At every step $t \in [1, T+1]$, the network predicts operator and memory attention vectors using Equation 9, 10,

and 11. The input is the `query` for $1 \leq t \leq T$ and a special `END` token when $t = T + 1$.

$$\mathbf{h_t} = \text{update} \left( \mathbf{h_{t-1}}, \text{input} \right) \tag{9}$$

$$\mathbf{a_t} = \text{softmax} \left( W \mathbf{h_t} + b \right) \tag{10}$$

$$\mathbf{b_t} = \text{softmax} \left( [\mathbf{h_0}, \dots, \mathbf{h_{t-1}}]^T \mathbf{h_t} \right) \tag{11}$$

The system then performs the computation in Equation 7 and stores $\mathbf{u_t}$ into the memory. The memory holds each step's partial inference results, i.e. $\{\mathbf{u_0}, \dots, \mathbf{u_t}, \dots, \mathbf{u_{T+1}}\}$. Figure 2 shows an overview of the system. The final inference result $\mathbf{u}$ is just the last vector in memory, i.e. $\mathbf{u_{T+1}}$. As discussed in Equation 4, the objective is to maximize $\mathbf{v}_\mathbf{y}^T \mathbf{u}$. In particular, we maximize $\log \mathbf{v}_\mathbf{y}^T \mathbf{u}$ because the nonlinearity empirically improves the optimization performance. We also observe that normalizing the memory vectors (i.e. $\mathbf{u_t}$) to have unit length sometimes improves the optimization.

To recover logical rules from the neural controller system, for each `query` we can write rules and their confidences $\{\alpha_l, \beta_l\}$ in terms of the attention vectors $\{\mathbf{a_t}, \mathbf{b_t}\}$. Based on the relationship between Equation 3 and Equation 6-8, we can recover rules by following Equation 7 and keep track of the coefficients in front of each matrix $\mathbf{M}_{\mathrm{R_k}}$. The detailed procedure is presented in Algorithm 1.

---

**Algorithm 1** Recover logical rules from attention vectors

---

    **Input:** attention vectors $\{\mathbf{a_t} \mid t = 1, \dots, T\}$ and $\{\mathbf{b_t} \mid t = 1, \dots, T + 1\}$
    **Notation:** Let $R_t = \{r_1, \dots, r_l\}$ be the set of partial rules at step $t$. Each rule $r_l$ is represented by a pair of $(\alpha, \ \beta)$ as described in Equation 1, where $\alpha$ is the confidence and $\beta$ is an ordered list of relation indexes.
    **Initialize:** $R_0 = \{r_0\}$ where $r_0 = (1, \ ( \ ))$.
    **for** $t \leftarrow 1$ to $T + 1$ **do**
        **Initialize:** $\widehat{R_t} = \emptyset$, a placeholder for storing intermediate results.
        **for** $\tau \leftarrow 0$ to $t - 1$ **do**
            **for** rule $(\alpha, \ \beta)$ in $R_\tau$ **do**
                Update $\alpha' \leftarrow \alpha \cdot b_t^\tau$. Store the updated rule $(\alpha', \ \beta)$ in $\widehat{R_t}$.
        **if** $t \leq T$ **then**
            **Initialize:** $R_t = \emptyset$
            **for** rule $(\alpha, \ \beta)$ in $\widehat{R_t}$ **do**
                **for** $k \leftarrow 1$ to $|\mathbf{R}|$ **do**
                    Update $\alpha' \leftarrow \alpha \cdot a_t^k$, $\beta' \leftarrow \beta$ append $k$. Add the updated rule $(\alpha', \ \beta')$ to $R_t$.
        **else**
            $R_t = \widehat{R_t}$
    **return** $R_{T+1}$

---

## 4   Experiments

To test the reasoning ability of Neural LP, we conduct experiments on statistical relation learning, grid path finding, knowledge base completion, and question answering against a knowledge base. For all the tasks, the data used in the experiment are divided into three files: *facts*, *train*, and *test*. The *facts* file is used as the knowledge base to construct TensorLog operators $\{\mathbf{M}_{\mathrm{R_k}} \mid \mathrm{R_k} \in \mathbf{R}\}$. The *train* and *test* files contain query examples `query(head, tail)`. Unlike in the case of learning embeddings, we do not require the entities in *train* and *test* to overlap, since our system learns rules that are entity independent.

Our system is implemented in TensorFlow and can be trained end-to-end using gradient methods. The recurrent neural network used in the neural controller is long short-term memory [9], and the hidden state dimension is $128$. The optimization algorithm we use is mini-batch ADAM [11] with batch size $64$ and learning rate initially set to $0.001$. The maximum number of training epochs is $10$, and validation sets are used for early stopping.

### 4.1   Statistical relation learning

We conduct experiments on two benchmark datasets [12] in statistical relation learning. The first dataset, Unified Medical Language System (UMLS), is from biomedicine. The entities are biomedical

concepts (e.g. `disease`, `antibiotic`) and relations are like `treats` and `diagnoses`. The second dataset, Kinship, contains kinship relationships among members of the Alyawarra tribe from Central Australia [6]. Datasets statistics are shown in Table 1. We randomly split the datasets into *facts, train, test* files as described above with ratio 6:2:1. The evaluation metric is Hits@10. Experiment results are shown in Table 2. Comparing with Iterative Structural Gradient (ISG) [27], Neural LP achieves better performance on both datasets. [2] We conjecture that this is mainly because of the optimization strategy used in Neural LP, which is *end-to-end* gradient-based, while ISG's optimization alternates between structure and parameter search.

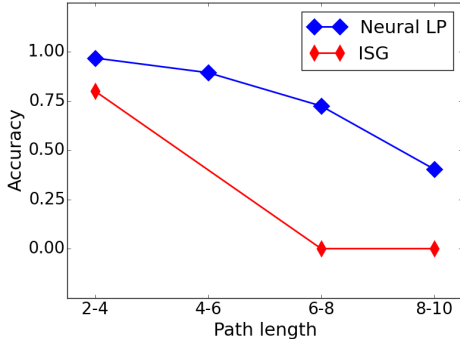

Figure 3: Accuracy on grid path finding.

Table 1: Datasets statistics.

|         | # Data | # Relation | # Entity |
|---------|--------|------------|----------|
| UMLS    | 5960   | 46         | 135      |
| Kinship | 9587   | 25         | 104      |

Table 2: Experiment results. $T$ indicates the maximum rule length.

|         | ISG | | Neural LP | |
|---------|---------|---------|---------|---------|
|         | $T=2$ | $T=3$ | $T=2$ | $T=3$ |
| UMLS    | 43.5 | 43.3 | 92.0 | **93.2** |
| Kinship | 59.2 | 59.0 | **90.2** | 90.1 |

## 4.2 Grid path finding

Since in the previous tasks the rules learned are of length at most three, we design a synthetic task to test if Neural LP can learn longer rules. The experiment setup includes a knowledge base that contains location information about a 16 by 16 grid, such as `North((1,2),(1,1))` and `SouthEast ((0,2),(1,1))` The `query` is randomly generated by combining a series of directions, such as `North_SouthWest`. The train and test examples are pairs of `start` and `end` locations, which are generated by randomly choosing a location on the grid and then following the queries. We classify the queries into four classes based on the path length (i.e. Hamming distance between `start` and `end`), ranging from two to ten. Figure 3 shows inference accuracy of this task for learning logical rules using ISG [27] and Neural LP. As the path length and learning difficulty increase, the results show that Neural LP can accurately learn rules of length 6-8 for this task, and is more robust than ISG in terms of handling longer rules.

## 4.3 Knowledge base completion

We also conduct experiments on the canonical knowledge base completion task as described in [3]. In this task, the `query` and `tail` are part of a missing data tuple, and the goal is to retrieve the related `head`. For example, if `HasOfficeInCountry(USA,Uber)` is missing from the knowledge base, then the goal is to reason over existing data tuples and retrieve `USA` when presented with query `HasOfficeInCountry` and `Uber`. To represent the `query` as a continuous input to the neural controller, we jointly learn an embedding lookup table for each `query`. The embedding has dimension 128 and is randomly initialized to unit norm vectors.

The knowledge bases in our experiments are from WordNet [17, 10] and Freebase [2]. We use the datasets WN18 and FB15K, which are introduced in [3]. We also considered a more challenging dataset, FB15KSelected [25], which is constructed by removing near-duplicate and inverse relations from FB15K. We use the same train/validation/test split as in prior work and augment data files with reversed data tuples, i.e. for each `relation`, we add its inverse `inv_relation`. In order to create a

*facts* file which will be used as the knowledge base, we further split the original train file into *facts* and *train* with ratio 3:1. [3] The dataset statistics are summarized in Table 3.

Table 3: Knowledge base completion datasets statistics.

| Dataset | # Facts | # Train | # Test | # Relation | # Entity |
|---|---|---|---|---|---|
| WN18 | 106,088 | 35,354 | 5,000 | 18 | 40,943 |
| FB15K | 362,538 | 120,604 | 59,071 | 1,345 | 14,951 |
| FB15KSelected | 204,087 | 68,028 | 20,466 | 237 | 14,541 |

The attention vector at each step is by default applied to all relations in the knowledge base. Sometimes this creates an unnecessarily large search space. In our experiment on FB15K, we use a subset of operators for each query. The subsets are chosen by including the top 128 relations that share common entities with the query. For all datasets, the max rule length $T$ is 2.

The evaluation metrics we use are Mean Reciprocal Rank (MRR) and Hits@10. MRR computes an average of the reciprocal rank of the desired entities. Hits@10 computes the percentage of how many desired entities are ranked among top ten. Following the protocol in Bordes et al. [3], we also use *filtered* rankings. We compare the performance of Neural LP with several models, summarized in Table 4.

Table 4: Knowledge base completion performance comparison. TransE [4] and Neural Tensor Network [24] results are extracted from [29]. Results on FB15KSelected are from [25].

| | WN18 | | FB15K | | FB15KSelected | |
|---|---|---|---|---|---|---|
| | MRR | Hits@10 | MRR | Hits@10 | MRR | Hits@10 |
| Neural Tensor Network | 0.53 | 66.1 | 0.25 | 41.4 | - | - |
| TransE | 0.38 | 90.9 | 0.32 | 53.9 | - | - |
| DISTMULT [29] | 0.83 | 94.2 | 0.35 | 57.7 | **0.25** | **40.8** |
| Node+LinkFeat [25] | 0.94 | 94.3 | **0.82** | 87.0 | 0.23 | 34.7 |
| Implicit ReasoNets [23] | - | **95.3** | - | **92.7** | - | - |
| Neural LP | **0.94** | 94.5 | 0.76 | 83.7 | 0.24 | 36.2 |

Neural LP gives state-of-the-art results on WN18, and results that are close to the state-of-the-art on FB15K. It has been noted [25] that many relations in WN18 and FB15K have inverse also defined, which makes them easy to learn. FB15KSelected is a more challenging dataset, and on it, Neural LP substantially improves the performance over Node+LinkFeat [25] and achieves similar performance as DISTMULT [29] in terms of MRR. We note that in FB15KSelected, since the test entities are rarely directly linked in the knowledge base, the models need to reason explicitly about compositions of relations. The logical rules learned by Neural LP can very naturally capture such compositions.

Examples of rules learned by Neural LP are shown in Table 5. The number in front each rule is the normalized confidence, which is computed by dividing by the maximum confidence of rules for each `relation`. From the examples we can see that Neural LP successfully combines structure learning and parameter learning. It not only induce multiple logical rules to capture the complex structure in the knowledge base, but also learn to distribute confidences on rules.

To demonstrate the *inductive* learning advantage of Neural LP, we conduct experiments where training and testing use disjoint sets of entities. To create such setting, we first randomly select a subset of the test tuples to be the test set. Secondly, we filter the train set by excluding any tuples that share entities with selected test tuples. Table 6 shows the experiment results in this inductive setting.

Table 5: Examples of logical rules learned by Neural LP on FB15KSelected. The letters `A`,`B`,`C` are ungrounded logic variables.

| | |
|---|---|
| 1.00 | `partially_contains(C,A) ← contains(B,A) ∧ contains(B,C)` |
| 0.45 | `partially_contains(C,A) ← contains(A,B) ∧ contains(B,C)` |
| 0.35 | `partially_contains(C,A) ← contains(C,B) ∧ contains(B,A)` |
| 1.00 | `marriage_location(C,A) ← nationality(C,B) ∧ contains(B,A)` |
| 0.35 | `marriage_location(B,A) ← nationality(B,A)` |
| 0.24 | `marriage_location(C,A) ← place_lived(C,B) ∧ contains(B,A)` |
| 1.00 | `film_edited_by(B,A) ← nominated_for(A,B)` |
| 0.20 | `film_edited_by(C,A) ← award_nominee(B,A) ∧ nominated_for(B,C)` |

Table 6: Inductive knowledge base completion. The metric is Hits@10.

| | WN18 | FB15K | FB15KSelected |
|---|---|---|---|
| TransE | 0.01 | 0.48 | 0.53 |
| Neural LP | **94.49** | **73.28** | **27.97** |

As expected, the inductive setting results in a huge decrease in performance for the TransE model[4], which uses a transductive learning approach; for all three datasets, Hits@10 drops to near zero, as one could expect. In contrast, Neural LP is much less affected by the amount of unseen entities and achieves performance at the same scale as the non-inductive setting. This emphasizes that our Neural LP model has the advantage of being able to transfer to unseen entities.

### 4.4 Question answering against knowledge base

We also conduct experiments on a knowledge reasoning task where the `query` is "partially structured", as the `query` is posed partially in natural language. An example of a partially structured query would be "in which country does x has an office" for a given entity x, instead of `HasOfficeInCountry(Y, x)`. Neural LP handles queries of this sort very naturally, since the input to the neural controller is a vector which can encode either a structured query or natural language text.

We use the WIKIMOVIES dataset from Miller et al. [16]. The dataset contains a knowledge base and question-answer pairs. Each question (i.e. the `query`) is about an entity and the answers are sets of entities in the knowledge base. There are 196,453 train examples and 10,000 test examples. The knowledge base has 43,230 movie related entities and nine relations. A subset of the dataset is shown in Table 7.

Table 7: A subset of the WIKIMOVIES dataset.

| | |
|---|---|
| Knowledge base | `directed_by(Blade Runner, Ridley Scott)` <br> `written_by(Blade Runner, Philip K. Dick)` <br> `starred_actors(Blade Runner, Harrison Ford)` <br> `starred_actors(Blade Runner, Sean Young)` |
| Questions | What year was the movie `Blade Runner` released? <br> Who is the writer of the film `Blade Runner`? |

We process the dataset to match the input format of Neural LP. For each question, we identity the `tail` entity by checking which words match entities in the knowledge base. We also filter the words in the question, keeping only the top 100 frequent words. The length of each question is limited to six words. To represent the `query` in natural language as a continuous input for the neural controller, we jointly learn a embedding lookup table for all words appearing in the `query`. The `query` representation is computed as the arithmetic mean of the embeddings of the words in it.

We compare Neural LP with several embedding based QA models. The main difference between these methods and ours is that Neural LP does not embed the knowledge base, but instead learns to compose operators defined on the knowledge base. The comparison is summarized in Table 8. Experiment results are extracted from Miller et al. [16].

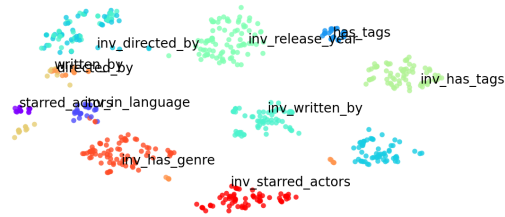

Table 8: Performance comparison. Memory Network is from [28]. QA system is from [4].

| Model | Accuracy |
|---|---|
| Memory Network | 78.5 |
| QA system | 93.5 |
| Key-Value Memory Network [16] | 93.9 |
| Neural LP | **94.6** |

Figure 4: Visualization of learned logical rules.

To visualize the learned model, we randomly sample 650 questions from the test dataset and compute the embeddings of each question. We use tSNE [15] to reduce the embeddings to the two dimensional space and plot them in Figure 4. Most learned logical rules consist of one relation from the knowledge base, and we use different colors to indicate the different relations and label some clusters by relation. The experiment results show that Neural LP can successfully handle queries that are posed in natural language by jointly learning word representations as well as the logical rules.

## 5 Conclusions

We present an end-to-end *differentiable* method for learning the parameters as well as the structure of logical rules for knowledge base reasoning. Our method, Neural LP, is inspired by a recent probabilistic differentiable logic TensorLog [5]. Empirically Neural LP improves performance on several knowledge base reasoning datasets. In the future, we plan to work on more problems where logical rules are essential and complementary to pattern recognition.

### Acknowledgments

This work was funded by NSF under IIS1250956 and by Google Research.

## Footnotes

[1]In this work, the notion of `query` refers to relations, which differs from conventional notion, where query usually contains relation and entity.

[2]We use the implementation of ISG available at `https://github.com/TeamCohen/ProPPR`. In Wang et al. [27], ISG is compared with other statistical relational learning methods in a different experiment setup, and ISG is superior to several methods including Markov Logic Networks [12].

[3] We also make minimal adjustment to ensure that all `query` relations in *test* appear at least once in *train* and all entities in *train* and *test* are also in *facts*. For FB15KSelected, we also ensure that entities in *train* are not directly linked in *facts*.

[4]We use the implementation of TransE available at `https://github.com/thunlp/KB2E`.

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
