[Reviews · NeurIPS 2017]

Reviewer 1



This paper develops a model for learning to answer queries in knowledge bases with incomplete data about relations between entities. For example, the running example in the paper is answering queries like HasOfficeInCountry(Uber, ?), when the relation is not directly present in the knowledge base, but supporting relations like HasOfficeInCity(Uber, NYC) and CityInCountry(NYC, USA). The aim in this work is to learn rules like HasOfficeInCountry(A, B) <= HasOfficeInCountry(A, C) && CityInCountry(C, B). Note that this is a bit different from learning embeddings for entities in a knowledge base, because the rule to be learned is abstract, not depending on any specific entities. The formulation in this paper is cast the problem as one of learning two components: - a set of rules, represented as a sequence of relations (those that appear in the RHS of the rule) - a real-valued confidence on the rule The approach to learning follows ideas from Neural Turing Machines and differentiable program synthesis, whereby the discrete problem is relaxed to a continuous problem by defining a model for executing the rules where all rules are executed at each step and then averaged together with weights given by the confidences. The state that is propagated from step to step is softmax distribution over entities (though there are some additional details to handle the case where rules are of different lengths). Given this interpretation, the paper then suggests that a recurrent neural network controller can be used to generate the continuous parameters needed for the execution. This is analogous to the neural network controller in Neural Turing Machines, Neural Random Access Machines, etc. After the propagation step, the model is scored based upon how much probability ends up on the ground truth answer to the query. The key novelty is developing two representations of the confidence-weighted rules (one which is straightforward to optimize as a continuous optimization problem, and one as confidence weights on a set of discrete rules) along with an algorithm for converting from the continuous-optimization-friendly version to the rules version. To my knowledge, this is a novel and interesting idea, which is a neat application of ideas from the neural programs space. Experimentally, the method performs very well relative to what looks to be a strong set of baselines. The experiments are thorough and thoughtful with a nice mix of qualitative and quantitative results, and the paper is clearly written. One point that I think could use more clarification is what the interpretation is of what the neural network is learning. It is not quite rules, because (if I understand correctly), the neural controller is effectively deciding which part of the rules to apply next, during the course of execution. This causes me a bit of confusion when looking at Table 3. Are these rules that were learned relative to some query? Or is there a way of extracting query-independent rules from the learned model?

Reviewer 2



This paper extends TensorLog to do rule learning. It transforms a weighted logic program into an approximate objective that is easy to optimize with backprop. The paper is rather clear and well-written. Although there are so many approximations, transformations and tricks involved that it may be hard to reproduce and fully understand what goes on. The selling point of the learner is that it is differentiable and interpretable. I'm not really convinced of either point. Yes, rules are interpretable, and even the objective of (2) is still quite intuitive and elegant. But once you flip sum and product (for no reason except that we can) in (5), any claim of interpretability becomes void. The recurrent attention model then goes further down the rabbit hole. Being differentiable is a trendy phrase, but in this context I wonder what it really means. When learning rules of length 2, one can just generate all possible rules and reduce the problem to parameter learning on this large rule set. And parameter learning on these (semi-)probabilistic models is always differentiable. For example, you can optimize the parameters of a set of MLN rules by differentiating the pseudolikelihoood. I don't find the argument of being differentiable particularly compelling or meaningful (unless I'm missing some important detail here). Can you tie the variables between relations in a rule? Seems like that is not possible to express? Such limitations should be discussed clearly. If the advantage compared to embeddings is that you can predict for unseen entities, why not set up a convincing experiment to prove that? Is this covered by any of the existing experiments? The experiments look good at first sight, but also appear to be somewhat cherry-picked to work with the learner. Rule lengths are 2-3 (why?), the limitations and approximations of the learner do not seem to matter on these tasks. Overall I don't extract any insight from the experiments; it's just a big win for the proposed learner with no explanation as to why or when not.

Reviewer 3



This paper suggest a rule-learning approach knowledge-base reasoning. Unlike traditional approaches that define rule-learning as a combinatorial search problem this paper proposes a differentiable model for learning both the rule structure and its parameterization for the knowledge base completion task. The learning problem is defined over an RNN. The approach was evaluated over three KB tasks, achieving state-of-the-art results. I found the paper difficult to follow, as in many cases the authors made claims that were either unexplained or out of context. Specifically, it would be helpful if the authors clarified in the related work section the differences and similarities to other neural program inductions frameworks (it's not clear how the example explains this difference). The transition from Eq.5 to Eq. 6-8 should also be explained more carefully. Terminology such as memory and attention should be explained before it is used. The equivalence between NL and rules in Sec. 4.3 was also not clear.